# Bone Regeneration: Mini-Review and Appealing Perspectives

**DOI:** 10.3390/bioengineering12010038

**Published:** 2025-01-07

**Authors:** Sylvain Le Grill, Fabien Brouillet, Christophe Drouet

**Affiliations:** 1CIRIMAT, Toulouse INP, Université Toulouse 3 Paul Sabatier, CNRS, Université de Toulouse, 4 Allée Emile Monso, BP44362, CEDEX 4, 31030 Toulouse, France; fabien.brouillet@univ-tlse3.fr (F.B.); christophe.drouet@toulouse-inp.fr (C.D.); 2Regenerative Nanomedicine Unit, Center of Research on Biomedicines of Strasbourg (CRBS), French National Institute of Health and Medical Research (INSERM), University of Strasbourg, UMR 1260, 1 Rue Eugène Boeckel, 67000 Strasbourg, France

**Keywords:** bone regeneration, bone substitute, hybrid, bioactive implants, tissue regeneration, implants, bone materials

## Abstract

Bone is a natural mineral-organic nanocomposite protecting internal organs and allowing mobility. Through the ages, numerous strategies have been developed for repairing bone defects and fixing fractures. Several generations of bone repair biomaterials have been proposed, either based on metals, ceramics, glasses, or polymers, depending on the clinical need, the maturity of technologies, and knowledge of the natural constitution of the bone tissue to be repaired. The global trend in bone implant research is shifting toward osteointegrative, bioactive and possibly stimuli-responsive biomaterials and, where possible, resorbable implants that actively promote the regeneration of natural bone tissue. In this mini-review, the fundamentals of bone healing materials and clinical challenges are summarized and commented on with regard to progressing scientific discoveries. The main types of bone-healing materials are then reviewed, and their specific relevance to the field is reminded, with the citation of reference works. In the final part, we highlight the promise of hybrid organic-inorganic bioactive materials and the ongoing research activities toward the development of multifunctional or stimuli-responsive implants. This contribution is expected to serve as a commented introduction to the ever-progressing field of bone regeneration and highlight trends of future-oriented research.

## 1. Introduction

From past civilizations to contemporary times, people have endlessly attempted to repair the human body with the view of restoring its functional aptitudes. This is especially true with regard to the reparation of the skeleton, as it allows mobility while protecting internal organs. Hence, along the ages, an evolution of bone repair materials has been on its way, motivated by the necessity to reestablish initial skeletal functions [1,2]. In this quest for skeletal repair, bone defects were tentatively filled with all sorts of materials, initially from natural origin, such as coral, nacre, cow bone, gold, wood, jade, etc. With the progress of technology and general knowledge of human metabolism, synthetic bone substitutes have progressively been set up, leading to prosthetic devices made of ceramic (initially non-resorbable, highly crystalline compounds) or metal (often in the form of alloys). Early on, however, the lack of continuity between the native bone and the implant was identified as one of the main limitations, potentially requiring the replacement of the implant due to mechanical loosening due to the absence of actual chemical bonding. Toward the 17th century, the relevance of long-lasting repair and improved osteointegration was assessed; for example, when Meekeren observed the bony fusion of the skull of an injured soldier with the dog skull fragment he used as a graft. This discovery paved the way for the subsequent development of biomaterials for bone integration [3]. Strategies were implemented to create a continuity between the implant and the surrounding bone. Hence, the integration of the graft was aimed to be stronger than that of the previous generation of prostheses. Among these strategies, the use of prosthesis coatings appeared as one leading pathway. For instance, the coverage with hydroxyapatite (HA) on the surface of prostheses made of TA6V alloy proved to allow enhanced biocompatibility and long-term stability [4]. The choice of HA for the composition of the coating was not trivial there, as it is the high-temperature ceramic that approaches the most natural composition of bone mineral. This strategy has been exploited since the 1980s to coat, especially in Europe, the femoral stem of hip implants, with significant improvement in the prosthesis durability in the body [5]. Further development of coating strategies and the use of alternative inorganic materials, e.g., octacalcium phosphate, was then carried out with the view of improving integration to the surrounding bone matrix [6].

In the constantly evolving domain of biomaterials for tissue engineering, a persistent quest for innovative materials, methodologies, or approaches can be outlined. One step forward in the pursuit of new bone repair biomaterials is the development of compounds capable of triggering cell activity. While autografts, allografts and xenografts are natural bone specimens and, therefore, *a priori* benefit from enhanced biointegration properties in comparison with synthetic materials (alloplastic bone substitutes), the latter can be finely engineered and tuned to be significantly enhanced. Also, they are virus-free and can be produced in large amounts and in reproducible batches. Moreover, in some cases, as for bioinspired calcium phosphates, their bioactivity can be tuned to adjust to clinical needs, for example, by being combined with therapeutic agents to provide new properties such as antibacterial, antitumoral, hemostatic, etc. [7,8,9]. A great deal of research activities are nowadays oriented toward the generation of such bioactive/multifunctional bone substitutes [10,11,12,13], which may thus overtake the natural bone-derivate grafts in terms of biointegration and bioactivity properties and can be tailored to release drugs. In this approach, one appealing concept is to take into consideration the factor of “time”, through the development of stimuli-responsive materials able to evolve upon external or internal stimuli. This “4D” strategy paves the way for so-called smart(er) materials, and Sajjad et al. and Javaid et al. have reported, for example, such 4D strategies, including those for tissue engineering [14,15].

Another appealing approach to developing bioactive bone repair materials that encompasses the time dimension is the development of resorbable materials. This strategy aims to induce the progressive neoformation of a bony matrix guided by the implanted biomaterial, which may be accompanied by the activation of bone cells in the context of actual “regenerative” medicine. Biomaterial resorption can also be responsive to stimuli [16,17,18], and several triggering factors in vivo may include temperature, pH, and biomolecular inputs from hormones, proteins [19], etc. This strategy, which is based on active resorbable biomaterials, ultimately aims to help the body rebuild tissues using natural processes. This approach minimizes the risk of loosening and, thus, the need to replace the implant prematurely. During the development of a biomaterial involving such a regenerative medicine approach, three main steps have to be considered: the properties and behavior of the material before the triggering effect, the physicochemical evolution of the material and the characteristics of the evolved material itself. The first step corresponds to the material prior to implantation and includes the investigation of non-toxicity, mechanical properties (allowing handling), storage and chemical binding that limit the risks of rejection. The second step mainly involves the colonization of the material by cells and the control of the implant degradation rate. The last step should mainly focus on the presence or absence of byproducts from the evolution of the material.

A brief introduction to bone chemistry, mechanical properties and cell activity will be provided in the first part of this review. We will present the complexity of this natural nanocomposite tissue and extract the critical properties that a bone substitute material should ideally exhibit to interact at best with the surrounding natural tissue. With this, a short review of “historical” bone healing strategies will be presented. In the second part of this review, our attention will more specifically focus on approaches implemented to reduce the need for further surgical operations to improve the biointegration of implants and/or their resorption. Finally, some perspectives will be given in the dynamic field of bone substitute materials.

## 2. Fundamentals of Bone Healing Materials: Definition, Needs and Expected Properties

In order to develop an appropriate bone substitute material, a deep understanding of the nature of the natural bone matrix is required. In Section 2, we will thus introduce the complexity of bone tissue. Indeed, bone is a multi-scale composite material that evolves with time and upon mechanical stress and host cells’ activity [20,21,22,23,24]. These chemical, physical, and biological aspects act together in a complex manner that provides the bone with the required properties, allowing for strength, mobility, and protection and hosting biological reactions such as hematopoiesis within the bone marrow.

There are currently various issues relating to bone-related surgical operations and the global recovery of patients. Although not exhaustive, one can mention in particular risks of implant loosening due to poor implant-tissue interaction, low mechanical properties, implant evolution/corrosion in vivo, risks of infections and necrosis if not enough vascularized, slow bone healing, etc. Various types of bone biomaterials have thus been explored—which is still ongoing—to address these points. Bone grafting material should thus ideally be developed considering these various aspects. In Section 2.1, we will introduce the complexity of the bone chemistry and structure, and in Section 2.2, we will present the clinical challenges that have to be met by a bone substitute material. This section will help define the relevant technical specifications that bone substitute materials should have, as will be developed in Section 3.

### 2.1. Bone Composition, Structure and Main Properties

Apprehending the complexity of bone chemistry and its evolution in vivo is necessary to develop the most relevant bone substitute materials since it is important to understand the qualities and specificities of the tissue to be repaired.

#### 2.1.1. Bone Composition

From a materials science point of view, bone is a mineral-organic nanocomposite. The mineral phase represents about 70 wt.% of bone matter and is mainly composed of nonstoichiometric (calcium and hydroxyl-deficient) carbonated apatite, exhibiting a rather low degree of crystallinity and involving nanosized constitutive crystals. The organic component, accounting for about 20 wt.%, is made of type I collagen and a myriad of other proteins in lower proportions. The final ca. 10 wt.% of bone consists of water. All of it is organized at a nanoscale up to the macro-scale [20,21,24,25,26].

##### Type I Collagen

Collagen is the most abundant protein in mammalian bodies. Type I collagen represents 90% of the bone organic matrix and provides the plastic properties of bone, while the mineral phase generates appealing elastic properties [24,27]. It is typically a protein of 1050-amino-acid long (with the most common motif sequences being *Gly-Pro-X* and *Gly-X-hydroPro,* where *X* is an amino acid other than *Gly*, *Pro* or *hydroPro*). At a molecular level, collagen forms long fibers with a triple helix structure of 300 nm long and 1.5 nm wide, implying three polypeptidic chains [28]. This triple helix organizes in fibrils with a 67 nm gap through the binding of two *Lys* amino acids; therefore, type I collagen presents a cholesteric liquid crystal phase. Because of the abundance of amine side chain functions, collagen is soluble in an acidic medium. The structure of the collagen matrix templates the precipitation and growth of the mineral apatitic phase during the process of mineralization of the bone [29].

##### Bone Mineral Phase

Mineralization of the organic collagenic template takes place in the gaps formed by the collagen fibrils. While the exact mechanism of mineral phase precipitation is still under debate, the chemistry of the final mineral phase is now well-known and identified as poorly crystallized, nonstoichiometric apatite (nHAp), whose degree of carbonation increases with bone aging [30]. While the direct nucleation and growth of an apatite phase cannot be totally ruled out, the most probable scenario is that it is formed upon hydrolysis of at least one initial intermediate phase. The first hypothesis considers amorphous calcium phosphate (ACP) as a hydrated precursor phase. This is supported by in vitro experiments where the formation of ACP can be noticed at early stages of precipitation, followed by its evolution into the nanocrystalline apatite, forming the “regular” bone mineral phase [31,32,33,34,35]. The other hypothesis involves another hydrated calcium phosphate precursor phase, namely octacalcium phosphate (OCP, in its triclinic form), which is also known to be hydrolyzable into nanocrystalline apatite [36,37,38]. This second hypothesis seems supported by some observations in vivo of spectroscopic vibration bands attributed to the presence of an OCP phase [39]. In addition, the evolution of OCP to nHAp is facilitated by a vicinity between the two crystallographic structures [40,41]. Nonetheless, it is expected that this transition from OCP to nHAp should occur via a dissolution/reprecipitation mechanism.

The structure of the mineral phase of bone, as mentioned above, is intimately associated with the pattern of the collagen fibrils [42]. The nHAp is indeed present as platelets of about 200 nm long and 100 nm wide (albeit these dimensions depend on the stage of aging of the apatite phase itself). At a larger scale, the mineral phase is organized in bone in two main structures: osteons and trabecula. These two structures are present in different parts of the bone and have different roles in terms of cell activity, vascularization, bone healing and properties [43,44].

#### 2.1.2. Bone Structure and Related Mechanical Properties

Bone structure and its mechanical properties are fundamental to understanding its functions and roles in the body. As mentioned above, bone is a composite material mainly composed of an organic phase (collagen) and a mineral one (poorly crystallized, carbonate, calcium deficient apatite), which overall provide both strength and flexibility. The collagen pattern creates a framework with interlaced collagen fibrils, which are mineralized with apatite nanocrystals that impart tensile strength and resilience [26]. This contributes to the bone’s compressive strength, allowing it to withstand substantial loads [22].

The bone structure is categorized into two distinct types: cortical (compact) bone and trabecular (spongy) bone, each with unique mechanical properties, functions and biological features.

Cortical bone forms the dense outer layer of bones and accounts for approximately 80% of the skeletal mass. Its tightly packed structure provides significant strength and rigidity, making it essential for load-bearing and protection (typically 50–150 MPa tensile strength, 130–180 MPa compressive strength) [24]. The dense arrangement of collagen fibers and mineral crystals in cortical bone ensures high resistance to bending and torsion, which is crucial for maintaining structural integrity.

Trabecular bone, on the other hand, is found at the extremity of long bones and within the interior of vertebrae. It has a porous, lattice-like structure consisting of a network of trabeculae (rod and plate-like elements) that align along lines of mechanical stress [44,45]. This design allows the trabecular bone to absorb and distribute loads efficiently, providing shock absorption and flexibility [43]. Despite being less dense than cortical bone, the trabecular architecture is highly adaptive and capable of remodeling in response to mechanical stimuli, which is vital for maintaining bone health and function [22].

The interplay between these two bone types ensures a balance of strength, flexibility, and lightness, enabling the skeleton to support the body, protect vital organs, and facilitate movement. Understanding the collagen pattern and the differences between trabecular and cortical bone is thus crucial for developing biomaterials that can effectively mimic or support natural bone functions.

#### 2.1.3. Bone Cells

Bones play a crucial role in the hematopoiesis process, which occurs in the bone marrow. Indeed, it hosts stem cells and a variety of blood cells. In addition to this key role, the bone also hosts the activity of cells dedicated to its functions and to its remodeling. Among them, osteoblasts, osteoclasts and osteocytes can be cited as major cell types. Briefly, osteoblasts are in charge of rebuilding the bone matrix; osteoclasts dissolve the existing bone (the balance of the two activities ensures the optimal renewal of the bone in the so-called bone remodeling process). Finally, osteocytes are mature osteoblast cells that get embedded in the bone matrix and ensure the long-lasting maintenance of the bone matrix properties [46]. For deeper insights on bone physiology, the authors recommend reading, for example, the review by Salhotra et al. [23].

### 2.2. Clinical Challenges in Bone Surgery

Whether for orthopedics, maxillofacial or cranial applications, the field of bone surgery encompasses two main domains: trauma fixation and gap filling. The former is predominantly achieved using metallic osteosynthesis material [47], although ceramic parts may also be employed under specific conditions [48]. The primary goal is to mechanically fix the bone pieces together and/or to the implant. Therefore, such physical fixation requires materials with high mechanical strength and durability. Hence, this fixation material often involves metallic (and sometimes sintered ceramic) screws as well [49,50]. However, fixation alone may not always be sufficient for the bone healing process to occur correctly. In such cases, a bone-filling material can be advantageously used to bridge the space between the implant and the native bone, for example, or to fill bone gaps. This kind of material can take the form of granules, membranes, cements and more complex structures like 3D pieces, which can be porous or not [51,52,53,54,55].

Bone surgery biomaterials can also present the advantage of improving implant osteointegration [56,57] and enhancing the natural healing process. To further improve these properties, they can be shaped with more bioactive and adaptable materials [58,59]. Research activities on spark plasma sintering (SPS), cold sintering, and, more recently, 3D printing have initiated this path [60,61,62,63]. Beyond improving osteointegration, these materials can be chemically tailored and designed to resist infections and address issues like tissue necrosis, which is particularly important for immune-deficient or aged patients, for example [64,65]. The bioactivity of the bone substitute material can also be enhanced to promote osteoinduction (implying the recruitment of bone cells and fostering their activity). This can be achieved by selecting intrinsically bioactive minerals (e.g., biomimetic apatites) and/or adding specific chemical compounds (bioactive ions, (bio)molecules, drugs) and/or through the modification of the implant surface/porosity features; indeed, some structural characteristics or surface roughness have proven enhanced osteointegration properties [66,67,68,69].

In conclusion, to address the challenges of bone surgery and develop an effective bone substitute, the biomaterial should ideally have mechanical characteristics that match the natural bone tissue to ensure proper implant positioning and stability. Lower mechanical properties may nevertheless sometimes be acceptable in non-load bearing surgical sites and/or when the bone defect is of small size. Significantly higher mechanical properties may, in contrast, not prove to be desirable as the implant may then concentrate the mechanical stress. In addition to mechanical properties considerations, an ideal bone substitute should also present chemically appealing properties to favor biointegration and osteoinduction to promote bone healing and actual tissue regeneration. In some cases, the biomaterial could also advantageously be presented in a moldable form (e.g., paste) to ensure the optimal filling of bone defects of complex shapes. Lastly, in addition to the biocompatibility/non-(cyto)toxicity of the initial biomaterial when implanted, consideration must be given to the degradation byproducts—metabolites—as they might subsequently cause an inflammatory response and represent a potential risk for the implant’s long-term stability [70,71].

## 3. Bone Healing Materials Through the Different Families of Materials

As mentioned above, to be considered an optimal bone substitute, a biomaterial should ideally meet (or come close to) the native physical, chemical and biological characteristics of natural bone. The best material and one of the first used in bone grafting history is, therefore, the bone itself: an autograft consists of harvesting a small portion of bone tissue from the patient somewhere other than the surgical site. The bone graft can be collected from the iliac crest, for instance, or from the fibula (a bone from the lower limb that only bears low mechanical stress and is not absolutely necessary for the patient) and placed in the desired location [72]. However, this method has some strong drawbacks as it requires two surgical procedures in the same patient that increase the risks of infection and post-operative complications, not to mention pain and discomfort. The use of bone from another patient (allograft) or from another animal species (xenograft) is possible but increases the risk of graft rejection. To overcome these limitations and to ensure the regularity of the supply of appropriate biomaterials, synthetic bone substitutes have thus been increasingly developed.

The development of such synthetic biomaterials implies the renouncement of several bone-mimicking properties, but the technical specification for synthetic bone substitutes follows the historical pathway of bone grafting. Thus, the first properties that must be addressed are the protection and mobility. This might come at the expense of the perfectly mimicking chemistry and/or biological integration of the bone substitute [73]. While chemical and biological integration is important, the initial focus is to ensure that the biomaterial provides adequate structural support and facilitates mobility. A second generation of bone graft materials was developed in order to enhance the osteointegration of the implant. This can be tailored with the use of a coating mimicking the native mineral phase or bone and/or via processes that lead to a porous structure resembling the porous structure of trabecular bone. Such a porous network may favor the bone/substitute entanglement to ultimately decrease the risk of loosening [74]. The third generation of bone substitutes targets the possibility of reactivating the bone activity in the surgical site with a genuine strategy of “regenerative medicine” that aims at providing bone cells with a scaffold and an environment that enhance their activity, to ultimately rebuild the bone at the place where the defect was previously present [75,76]. Thus, the development of bone substitute materials has witnessed the emergence of a variety of synthetic or natural materials, including degradable and non-degradable materials. Non-degradable metals like 316L stainless steel and Ti-6Al-4V are widely used for load-bearing implants, while degradable metals like magnesium are favored for scaffold applications. Similarly, degradable polymers such as polylactic acid (PLA) and polyglycolic acid (PGA) are ideal for scaffolds, whereas non-degradable polymers like poly(aryl-ether-ether-ketone) (PEEK) may also be used for load-bearing use. Figure 1 provides an overview of the different families of bone substitute materials, each material addressing specific mechanical and biological requirements for bone repair in accordance with clinical needs.

Section 3 will thus present these three generations of bone substitute materials through a variety of materials, including nature and chemistry, with a highlight on the last one due to its more promising outcomes and future-oriented research.

### 3.1. Inorganic-Based Bone Implants

#### 3.1.1. Metal-Based Biomaterials

The most iconic bone implants are undeniably hip and knee prostheses, along with dental implants. Historically, these implants were developed using metallic materials, often as alloys such as titanium-based and, more recently, magnesium-based alloys [77,78] and stainless steel, typically 316L [79,80]. However, these materials exhibit mechanical properties that diverge from the natural bone matrix, which can lead to stress shielding at the interface, thus leading to possible prosthesis failure [81]. To counteract the mismatch between the mechanical properties of the metallic implant and the biomechanical properties of the bone, alloy materials can be designed with a controlled porosity and/or an increased surface roughness. These strategies are intended to increase bone-bonding ability in the implant. In fact, shaping porosities in an implant allows for reducing its Young modulus and promotes the ingrowth of new bone tissue that enhances “mechanical interlocking”. Among the settings influencing this interlocking and bone ingrowth are the total porosity [82] and the pore shape [83], and, more specifically, the angular orientation formed by the pores [67]. However, as presented above, the porosity of natural bone is not highly periodic but rather random. To approach such features, Zeng et al. [84] produced a fused-pore-shaped scaffold and demonstrated the reduction of the stress shielding effect, thus limiting the risks of aseptic loosening and prosthesis failure. Nonetheless, by modifying the shape of the alloy, its surface and corrosion properties are modified. Corrosion behavior has to be carefully monitored, as metallic ions and particles generated from implant corrosion can have a negative effect on the cells [79,85,86]. Costa et al., for example, demonstrated the good properties of Ti alloy (Ti-6Al-4V) with regard to the fretting corrosion between a Ti bulk and a Ti porous material in a biological medium [87]. Finally, an important parameter to consider, if pores are included in the shape of the implant, is the possible presence of micromotion [88]. In fact, the presence of pores in the structure reduces the stiffness of the material, which may consequently induce micromotions at the bone/implant interface. These micromotions can then allow the proliferation of fibrous connective tissue, inhibiting bone growth.

#### 3.1.2. Ceramics

Apart from metal-based implants, bone and dental substitutes are widely made of ceramics. The latter has become increasingly important in the field of biomaterials, particularly in applications related to bone repair. Zirconia (zirconium dioxide, ZrO_2_) is one of these representatives. This ceramic is mechanically highly resistant and can withstand important loads, promoting its use for orthopedic or dental implants. Furthermore, its natural aesthetic white color encourages its use in the dental domain [89]. In addition, zirconia is chemically stable: this last property is both an advantage as it allows for long-standing implant and a disadvantage as it restrains the binding of the implant with the surrounding environment. In addition, zirconia undergoes low-temperature degradation (a phase transition from quadratic to monoclinic), which lowers its mechanical properties and fracture propagation resistance and, therefore, its long-term application [90]. To overcome this limitation, surface modifications have been proposed to improve osteointegration [91] (i.e., the mechanical binding and fixation of the implant with the surrounding tissue) and osteoinduction [92] (i.e., the promotion of the surrounding bone growth and healing toward the implant). The latter modification can be performed by implementing a calcium phosphate-based (CaP) ceramic coating, as described by Cheng et al. [92]. In fact, this family of CaP ceramics presents the advantages of approaching the chemistry of the natural bone mineral phase.

For this reason, CaPs are often used either as coatings or as bulk (i.e., self-supported) materials. However, depending on the CaP phase considered, the crystalline nature of phases like stoichiometric HA may lead to lower biointegration and resorption rates compared to the amorphous (ACPs) or less crystalline (biomimetic apatites) counterparts [93,94,95]. Their unique properties, including biocompatibility, bioactivity, mechanical strength, and the ability to support bone growth, make them ideal candidates for medical applications. Ceramic materials made of HA and tricalcium phosphates (TCP, in the α or β forms) provide osteoconductive properties relevant to bone repair [93]. HA is one of the most widely used ceramics in bone substitutes [96]. In addition, Zhang et al. [97] have demonstrated the importance of the shape of the HA implant structure in favor of osteointegration and material resorption, leading to an even more tunable material. Additionally, HA can be used in various forms, including powders, granules, and blocks, to fill bone defects and voids of various types. Moreover, the possibility of shaping the HA “high-temperature” phase into porous structures may support vascularization and favor cell migration, which are essential features for effective bone repair. However, pure hydroxyapatite, as a ceramic material, remains somewhat brittle and may not be suitable for load-bearing applications, necessitating the combination with other materials to enhance its mechanical properties [59]. Biological features can also be improved. Bornert et al. [98], for instance, formulated an FDA-approved biphasic bone substitute with pro-angiogenic complexes to significantly improve vascularization in the long run, even for large defects.

Tricalcium phosphates (TCP) are other prominent ceramics used in bone repair. Available in two crystalline forms, alpha (α-TCP) and beta (β-TCP). Like some other CaPs, such as ACP or OCP, α-TCP can hydrolyze into bone-like apatite under physiological conditions. On the other hand, β-TCP can be resorbed upon cell activity [99]. This resorption property is particularly beneficial in applications where temporary scaffolding is required to support bone growth before being resorbed. β-TCP, in particular, is favored for its balance between resorption rate and mechanical stability. TCP is often used in composite forms, combined with other materials such as collagen, biodegradable polymers and other ceramic phases [100], to tailor its properties to specific clinical needs [101]. This adaptability makes TCP a versatile option for various bone grafting scenarios, from small defects to larger reconstructions. The most known example of combinations involving β-TCP is in the well-known HA/β-TCP biphasic calcium phosphate (BCP) parts, largely used nowadays in the clinic and where the presence of the β-TCP phase “artificially” confers some degradation behavior to such scaffolds involving the non-resorbable HA counterpart.

Besides high-temperature CaP phases such as HA, α-TCP, β-TCP, or even monetite, calcium phosphate ceramic-based biomaterials also include other compounds obtained at low temperatures such as nanocrystalline nonstoichiometric apatites (nHAp, so-called “biomimetic”)—closer to the natural bone mineral—as well as brushite, octacalcium phosphate (OCP) and amorphous calcium phosphates (ACPs). Biomimetic apatites, in particular, share the same (sub)structural and compositional features as bone minerals. In addition to being constituted of nanosized crystals, thus generating a large exchange surface with the surrounding medium, they are calcium- and hydroxyl-deficient, which leads to lower thermodynamic stability, allowing for increased bioresorption capacities. Moreover, and most importantly, the surface of biomimetic apatite nanocrystals—like natural bone apatite—is covered by a non-apatitic amorphous-like surface layer, with very labile (i.e., easily exchangeable) constituting surface ions. This specificity, as for natural bone, may then convey a wealth of biological properties for bone regeneration applications, such as improved osteogenic, angiogenic and healing properties as well as antimicrobial, anticancer, hemostatic or anti-osteoporotic features, depending on the nature of ions and/or of adsorbed molecular species (enzymes, drugs, plant-derived phytotherapeutics) on the nanocrystals [102,103,104].

Among the CaP phases that are increasingly investigated in the bone substitute field, besides biomimetic apatites, are OCP and ACPs. Indeed, they are both suspected to be involved in the early stages of the natural mineralization process occurring in bones in vivo, as mentioned previously. OCP also presents a noticeable affinity for several drugs and organic compounds. As OCP evolves, in physiological conditions, toward nanocrystalline apatite mimicking the bone mineral phase, it is a material of increasing interest in the bone regeneration field [36,37,105,106]. Similarly, ACPs have also witnessed an increasing interest due to a high metastability that may rapidly evolve toward bone-like apatite in aqueous medium. It is noteworthy here that there is not only one type of ACP compound but also a whole subfamily of CaPs. Indeed, while the “usual” ACP composition is the tricalcium form Ca_3_(PO_4_)_2_. nH_2_O, corresponding to a Ca/P molar ratio of 1.5, internal hydrolysis of PO_4_^3−^ ions into HPO_4_^2−^ and OH^−^ can also be carried out to yield a series of compounds of refined behaviors [107,108,109]. Plus, it is also possible to generate ACPs with Ca/P ratios different from 1.5, as was recently reported by the authors [110].

It is important to note that, contrary to high-temperature CaP phases, which can be processed/shaped by “conventional” ceramic/sintering methods, the hydrated/metastable CaP phases, such as biomimetic apatites, OCP and ACPs, cannot be treated with such harsh approaches. In this case, and with the view to exploit at maximum their high bioactivity compared to their less-reactive high-temperature counterparts, it is necessary to develop alternative, non-conventional ceramic approaches to produce cohesive 3D implants. In this view, low-temperature consolidation/cold sintering strategies have started to be applied by Drouet et al. starting in 2006 and later further extended—including by other authors and for other materials such as Mg/CO_3_-stabilized ACP—unveiling the possibility to yield 3D scaffolds for bone regeneration purposes. Further recent works from Le Grill et al. and Rubenis et al., in particular, have been conducted to explore non-conventional cold sintering ways to consolidate ACP without triggering its evolution prior to the implantation [61,62,111,112,113,114,115].

#### 3.1.3. Bioactive Glasses

Finally, bioactive glasses represent another important resource for bone repair applications. These materials are sometimes difficult to classify as they are sometimes assimilated to the vast class of oxide-based “ceramics”, while they may also be considered as a separate family of compounds.

Bioactive glasses, typically composed of silica (SiO_2_), calcium oxide (CaO), and phosphorous pentoxide (P_2_O_5_), are expected to bond with bone and stimulate cellular activity through the release of calcium or phosphate ions, as well as various potential doping ions that may, as for apatites, confer additional properties. Indeed, bioactive glasses can release ions, such as calcium, silicon or additional customized ions [116], which have been shown to enhance osteoblast proliferation and differentiation, further accelerating the bone healing process. Their ability to stimulate angiogenesis the formation of new blood vessels, is another crucial advantage, ensuring the newly formed bone is well-nourished and integrated. It is worth mentioning, however, that, upon implantation, bioactive glasses undergo surface reactions that result in the formation of a layer of carbonated apatite on the glass particle surface, which closely resembles the mineral phase of bone [117,118,119]. This layer not only facilitates a strong bonding with the surrounding bone tissue but also promotes osteogenesis, i.e., the formation of new bone. Note that bioactive glasses can also be associated with organic molecular species to provide additional biological functions, and the advent of phytotherapeutics may be listed for CaPs as an appealing additional option [120].

#### 3.1.4. Non-Oxide Ceramics

Non-oxide ceramics may also be encountered, although less frequently, in the bone repair field. They usually exhibit high strength, fracture toughness, excellent wear resistance and chemical stability [121]. Among them, silicon nitride (SiN) has emerged as a promising material in the field of bone substitutes. In particular, silicon nitride presents high mechanical strength, making it suitable for load-bearing applications, while its biocompatibility and ability to promote bone cell activity enhance osteoconductivity [122,123]. In addition, SiN was reported to have an intrinsic antibacterial effect due to its surface chemistry, which inhibits bacterial biofilm formation, hence limiting the risk of infection [123,124]. Finally, the chemical stability of SiN further ensures a long-term performance in a physiological environment, which may be relevant for some clinical applications. One of the limitations is, however, the cost of fabrication of such nonoxide ceramics. Also, they are rather complex to produce, which limits their spreading as bone substitute alternatives.

### 3.2. Organic-Including Materials

Softer materials have also been developed for bone implants. Polymers and hydrogels have significantly emerged here as pivot players in the field of biomaterials, allowing for the tunability of the inorganic-based material by enhancing their biocompatibility, incorporating an organic counterpart that can mimic the organic portion of the bone. These organic materials offer unique properties such as flexibility, biocompatibility, and the ability to be engineered and processed for specific purposes, making them highly suitable for medical use. Polymers, both natural and synthetic, including hydrogels, which are three-dimensional networks of hydrophilic polymers, have, therefore, shown immense potential in supporting bone regeneration and healing.

#### 3.2.1. Synthetic Polymers

One of the primary advantages of polymers in bone substitutes is their versatility. Synthetic polymers such as polylactic acid (PLA), polyglycolic acid (PGA), or their combinations as polylactic-co-glycolic acid (PLGA), as well as polycaprolactone (PCL), are widely used due to their degradability, biocompatibility, and mechanical properties. These materials can be engineered to degrade at controlled rates, matching the pace of new bone formation. For example, PLA and PGA can be combined to form PLGA, a copolymer that degrades into lactic and glycolic acids, which are naturally metabolized by the body [125]. This degradation process eliminates the need for a second surgery to remove the scaffold, reducing the patient’s overall treatment burden. Smaida et al. [126], for example, developed a polycaprolactone-HA composite associated with bone marrow cells that resorbed in four weeks. This material has proven to have the potential to increase mineralization and differentiation into osteoblasts. Improving bone regeneration.

Among synthetic polymers, poly(aryl-ether-ether-ketone) (PEEK) implants have attracted much attention, for example, in cranial, maxillofacial or orthopedic surgeries, e.g., as fusion cages for intervertebral fusion. PEEK is a chemically stable polymer under physiological conditions, with mechanical properties matching those of bone, thus making it suitable for load-bearing applications [127]. PEEK-based composites, for example, reinforced with carbon fibers and graphene oxide, can also allow increasing mechanical strength to reach clinical requirements. However, PEEK is inherently bioinert. While bioinertness can be advantageous in certain contexts, it limits cell adhesion and increases the risk of fibrous tissue formation around the implant, which may lead to loosening or implant failure. To address these limitations, trials have been conducted to chemically modify the surface of PEEK implants [128,129], as well as to introduce structural modifications [130]. In addition, PEEK has also been associated with inorganic phases to improve its biological response either through coating treatment [131] or by mixing it with materials such as SiN or CaP [132,133]. These modifications aim to enhance PEEKs’ integration with the surrounding tissue and cells and reduce the risk of complications. In contrast to PLA and PGA, which are designed for temporary applications, PEEK-based materials aim to function as long-term implants. Research is still ongoing to increase its bioactivity and osteointegration capacities.

#### 3.2.2. Natural Polymers

Natural polymers such as collagen, chitosan, alginate and hyaluronic acid also play crucial roles in the domain of bone substitutes. Collagen, the main protein in the extracellular matrix of bone and other human tissues, provides a natural support for cell attachment, proliferation, and differentiation [26,59]. Its inherent biocompatibility and biodegradability make it an excellent choice for bone substitute applications. In combination with other compounds such as CaP, it can help to promote osteogenesis in the first more risky weeks after surgery. Besides, Lin et al. [75] developed a surface modification layer-by-layer to enhance the Ti surface reactivity. The substrate was immersed in a NaOH solution in order to modify the surface chemistry, which in turn can bind with organic compounds such as collagen or chitosan.

Chitosan, derived from chitin, has antimicrobial properties in addition to its biocompatibility and biodegradability, making it useful for preventing infections in bone grafting procedures [134]. Therefore, it has been associated with CaP phases to enhance implants’ biological response and improve mechanical properties [135,136].

Hyaluronic acid, known for its hydrating properties and its lubricant role in the joints, can support cell migration and proliferation, facilitating the healing process [137]. It can be modified with methacrylate to shape it as a hydrogel. Hsia et al. demonstrated that methacrylated hyaluronic acid containing recombinant human amyloid protein could successfully promote osteogenic differentiation and improve bone regeneration in vivo [138].

#### 3.2.3. Hydrogels

Hydrogels, with their high water content and ability to mimic the natural extracellular matrix, are particularly promising for bone regeneration purposes. These materials can be engineered to encapsulate cells and growth factors, creating an environment conducive to bone regeneration [139,140]. Hydrogels can be injected into bone defects, where they conform to the shape of the defect and provide a scaffold that promotes tissue growth [141]. In this view, this has to be limited to small bone defects and/or non-load-bearing sites. The porous internal organization of hydrogels also facilitates nutrient and oxygen diffusion, which is essential for cell survival and function.

One notable example of hydrogels in bone grafting is the use of injectable hydrogels that can solidify in situ [142]. These hydrogels can be loaded with bioactive molecules such as bone morphogenetic proteins (BMPs), which have a strong effect on inducing bone formation [143]. The injectable nature of these hydrogels allows for minimally invasive procedures, reducing patients’ recovery time and surgical risks. Moreover, hydrogels can be designed to degrade at a rate that matches the rate of new tissue formation, ensuring that the scaffold supports the healing process without hindering it.

### 3.3. Composites and Hybrid Biomaterials

Organic and inorganic components can often be considered complementary in their physical, chemical, mechanical and biological properties. As such, they may thus be judiciously combined to yield improved 3D scaffolds and implants for bone regeneration, especially when both the organic and inorganic subparts exhibit bioactive features.

Polymers and hydrogels can thus, for example, be combined with other biomaterials to enhance their properties. For instance, composite materials that incorporate bioactive ceramics like hydroxyapatite (HA) or tricalcium phosphate (TCP) with polymers can provide the mechanical strength from ceramic part and the flexibility of the polymer counterpart. Such composites can offer a balanced approach, supporting both the mechanical and biological aspects of bone healing. The addition of HA or TCP to polymer matrices can enhance osteoconduction, promoting new bone growth directly on the scaffold [126,144]. The combination of natural polymers such as collagen with HA along with Bone Morphogenetic Protein (BMP) has also demonstrated enhanced osteoblast activity with respect to the unfunctionalized BMP material [145]. In the same study, the authors also suggested adding a strontium-enriched ACP phase. However, their results only pointed out a minor effect of this addition in comparison with the BMP-enriched material. Zheng et al. combined the injectability property of a hydrogel with the biological properties of bioactive glass to develop an injectable silk fibroin/mesoporous bioglass/sodium alginate composite promoting vascularization and bone regeneration into the bone defect [141]. In another context, Xu et al. developed a composite material designed for the loss of mechanical properties induced by Type II diabetes mellitus. This material was composed of a Sr doped-mesoporous bioactive glass embedded in a gelatin methacrylate matrix, with the view to act in the different steps of bone formation and remodeling by influencing the quality of the cross-linking of the collagen, thus promoting cell adhesion and production of non-collagenous proteins. This, combined with the Sr^2+^ ion effect of enhancing osteogenic differentiation and the bioactive glass propensity to induce calcium phosphate nucleation and growth, was found to lead to a cascade of events resulting in the formation of a healthier and more mechanical relevant bony matrix [146].

Advanced manufacturing techniques such as 3D printing have further expanded the possibilities of using polymers and hydrogels in bone grafting. Three-dimensional printing allows for the precise fabrication of scaffolds with complex geometries tailored to patient-specific needs [147,148]. By using bioprinting technologies, it is possible to incorporate cells, growth factors, and bioactive materials directly into the scaffold, creating a more effective and personalized treatment option [149]. This level of customization can improve the integration and functionality of the graft, leading to better clinical outcomes.

Another exciting development linked to the use of polymers and hydrogels in the constitution of bone substitutes is the creation of smart materials that respond to local stimuli. These materials can be designed, for example, to release therapeutic agents in response to specific triggers such as changes in pH, temperature, the use of (NIR) radiation, or else the presence of enzymes [150,151]. This targeted delivery can enhance the efficacy of bone healing treatments by providing a localized and sustained release of drug(s), growth factors, or other bioactive molecules [152].

Despite promising advances, there are still challenges to be addressed in the use of polymers and hydrogels in bone substitutes. Ensuring the long-term stability and integration of these components with host tissues remains a critical area of research. Additionally, optimizing the mechanical properties of these materials to match those of natural bone, particularly in load-bearing applications, is essential for their widespread adoption. This is intimately linked to combinations with an inorganic phase such as biomimetic apatites, ACPs or bioactive glasses, which can not only modulate the mechanical properties of the implants but also provide additional functionalities through their doping ions or adsorbed molecular species, as mentioned above. Furthermore, the immune response to these organic-based- or organic-inorganic composite materials needs careful consideration to avoid adverse reactions, including in the long term, and ensure successful outcomes.

In summary, the use of bone biomaterial components based on polymers and hydrogels holds significant potential, offering a range of properties that can be tailored to meet specific clinical needs, generally in association with an inorganic counterpart for complementary effects. Their ability to support cell growth, deliver bioactive molecules, and degrade in a controlled manner makes them ideal candidates for bone regeneration applications. Ongoing research and technological progress are likely to further enhance the capabilities of these materials, leading to even more effective and personalized bone substitutes. As these materials continue to evolve, they promise to play a crucial role in improving patient outcomes in bone repair and regeneration. It is possible to distinguish two categories of mineral-organic associations: while “composites” are essentially the addition of two components with no specific chemical interaction, “hybrids” can be the denomination reserved to systems in which each component interacts with the other at a molecular scale. In the latter case, for example, Raman, FTIR or NMR spectral features are detectably modified, rather than providing the simple addition of spectra. Such hybrids can, for instance, be obtained when precipitating the mineral phase in the presence of the organic moiety, as opposed to mechanical mixtures of the two components, which instead gives rise to a composite/multi-materials system. The conception and fabrication of mineral-organic hybrid biomaterials, compared to composites, could also provide a future-oriented appealing area of research, with the view to developing chemical links at the molecular scale, as in natural bone tissue.

## 4. Conclusions and Perspectives

The development of new bone implant materials can take advantage of various families of materials (metals, ceramics, glasses and polymers) to the benefit of patients and combine their respective intrinsic properties. Bone repair has witnessed an evolution from the reconstruction of the primal functions—protection and mobility—to the development of resorbable and bioactive materials that enhance the healing process kinetics and limit surgical risks. This evolution is illustrated in Figure 2. Furthermore, this development testifies to the fine interlocking of chemistry, physics, and biology. Chemistry should, as far as possible, play a central role in bone-implant interfaces, which can imply surface functionalization, e.g., exposing selected chemical groups to enhance biocompatibility, protein coverage and cell adhesion. Physics contributes to optimizing the structural aspects, ensuring that the graft possesses the necessary mechanical strength and porosity to support new bone growth and allow neo-vascularization. Biological properties are crucial for recruiting cells toward the implant and triggering effective bone regeneration. This multidisciplinary approach allows for the creation of advanced biomaterials that may not only mimic the natural bone specificities but also actively participate in the healing process, leading to improved clinical outcomes and faster patient recovery. The synergy between these fields continues to drive innovation, opening new avenues for personalized and effective bone grafting solutions.

Improvements are yet to come to lower the risks of rejection and reduce the need for additional surgery. To this aim, biomaterials research is clearly more and more oriented toward the use of bioactive compounds, either of inorganic nature such as biomimetic nanocrystalline apatites (very close to the mineral part of bones) or other reactive CaPs such as ACPs and OCP, or else biopolymers/hydrogels and their combinations together and with active agents to modulate the biological properties of such implants of the last generation. Here, stimuli-responsive and/or multifunctional biomaterials will be particularly relevant to study. Recent advances have also witnessed the increasing use of cells in the development of biomaterials. The next step forward in this domain might be to develop tissue-like bio-organic-inorganic materials with, for instance, the use of organoids or patient-derived tissues in order to further enhance the compatibility of the patient and the implant and take advantage of the bioactive molecules secreted by cells. Chemistry, physics, and biology will thus need to be combined and synergistically exploited, considering the clinical needs in terms of bone site and patients’ condition.

## Figures and Tables

**Figure 1 bioengineering-12-00038-f001:**
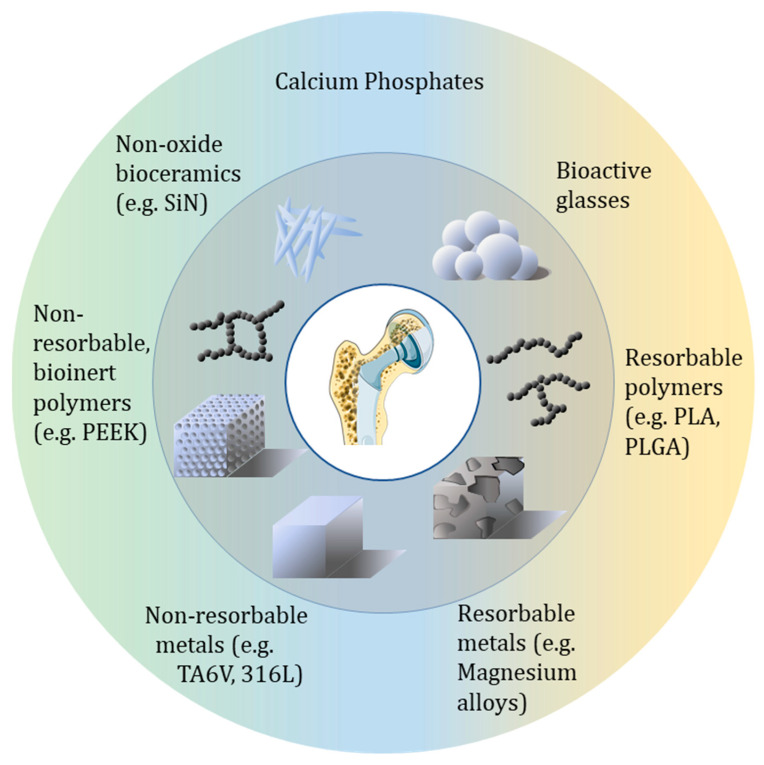
Overview of the different bone substitute materials categorized by their chemistry, their resorbable or non-resorbable behaviors and shaping possibilities: metals (316L, Ti-6Al-4V, magnesium alloys) as bulks solids or porous structures; ceramics (Calcium Phosphate, SiN) as scaffolds, coatings or granules; bioactive glasses as scaffolds or granules; polymers (PEEK, PLA, PLGA) as scaffolds, coatings or injectable forms.

**Figure 2 bioengineering-12-00038-f002:**
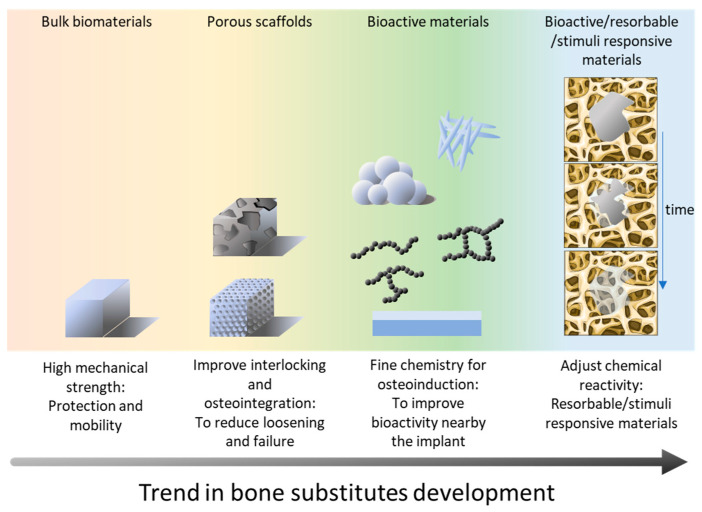
General evolution in the development of bone substitutes, from the initial need for protection and restoration of mobility to the latest bone substitute materials that are bioactive and promote bone regeneration by activating and stimulating cell activity in the surrounding of the implant.

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
