# Peer review of "Bone Regeneration: Mini-Review and Appealing Perspectives"

_bioengineering, 2025, doi:10.3390/bioengineering12010038_

Round 1

Reviewer 1 Report

Comments and Suggestions for Authors

This manuscript provided a short review for bone repair biomaterials including metals, polymers, ceramics and composites. The bone structure, mechanical properties and cell activity were described firstly in section 2.  The text in this section was well established in the literature.  Bone healing biomaterials as aforementioned were then introduced in section 3.  I found several important points were missing in section 3 including:

(1) Bone repair  biomaterials include degradable and non-degradable. Metal-based alloys e.g. 316L and Ti-6Al-4V are mainly  non-degradable with the exception of  magnesium. The former is  primarily used for load-bearing implants whereas the latter is mainly for scaffold applications. Similarly,  degradable polymers such as natural polymers, synthetic PLA, PGA etc. are for scaffold applications, and non-degradable PEEK composites are designed for load-bearing applications. The authors should mention these just under section 3.1.

(2) Subsection 3.1.1.

Ti-based and Mg-based alloys and 316L were briefly described as biomaterials for prostheses and dental implants.  However, these materials suffered from stress shielding effect especially for 316L with an elastic modulus of about 190 GPa and above.  Elastic modulus of Ti-6Al-4V alloy was lower than that of 316L, but still significantly higher than that of cortical bone of 6-30GPa. The authors should include these factors and related references in the revised manuscript.

(3) Subsection 3.3. Composites and hybrid biomaterials.

The challenges for making polymer-based composites with mechanical properties matching with human bones, particularly for load-bearing applications were addressed. However, the types of polymer composites that suitable for these applications were not mentioned. It is known that the mechanical properties of most polymers are much lower than those of human bones. However, PEEK-based composites, especially reinforced with carbon fibers, and even graphene oxide  can meet those mechanical requirements. The authors should mention and add related references in their revised paper. 

Author Response

Comments 1: Several important points were missing in section 3 including:

(1) Bone repair biomaterials include degradable and non-degradable. Metal-based alloys e.g. 316L and Ti-6Al-4V are mainly on-degradable with the exception of magnesium. The former is primarily used for load-bearing implants whereas the latter is mainly for scaffold applications. Similarly, degradable polymers such as natural polymers, synthetic PLA, PGA etc. are for scaffold applications, and non-degradable PEEK composites are designed for load-bearing applications. The authors should mention these just under section 3.1.

Answer: We thank the reviewer for this comment. This rapid overview of the bone substitute materials has been implemented in the text. However, it has been added prior the section 3.1 at the end of the paragraph 3.

Improvements in the text: Line 283:

Thus, the development of bone substitute materials has witnessed the emergence of a variety of synthetic or natural materials including degradable and non-degradable materials. Non-degradable metals like 316L stainless steel and Ti-6Al-4V are widely used for load-bearing implants, while degradable metals like magnesium are favored for scaffold applications. Similarly, degradable polymers such as polylactic acid (PLA) and polyglycolic acid (PGA) are ideal for scaffolds, whereas non-degradable polymers like poly(aryl-ether-ether-ketone) (PEEK) may also be used for load-bearing use. Each material addresses specific mechanical and biological requirements for bone repair in accordance with clinical needs.

Line 292:

The present section 3 will thus present these three generations of bone substitute materials through a variety of materials nature and chemistry, with a highlight to the last one due to its more promising outcomes and future-oriented research.

Comments 2:

(2) Subsection 3.1.1.

Ti-based and Mg-based alloys and 316L were briefly described as biomaterials for prostheses and dental implants. However, these materials suffered from stress shielding effect especially for 316L with an elastic modulus of about 190 GPa and above. Elastic modulus of Ti-6Al-4V alloy was lower than that of 316L, but still significantly higher than that of cortical bone of 6-30GPa. The authors should include these factors and related references in the revised manuscript.

Answer: We thank the reviewer for this comment. However, this point has already been addressed in the main text as mentioned in Line 291 “However, these materials exhibit mechanical properties that diverge from the natural bone matrix that can leads to stress shielding at the interface, thus to possible prosthesis failure [81].” Strategies to lower the elastic modulus are also mentioned (line 299 : “Among the settings influencing this interlocking and bone ingrowth are the total porosity [82] and the pore shape [83].”).

Comments 3:

(3) Subsection 3.3. Composites and hybrid biomaterials.

The challenges for making polymer-based composites with mechanical properties matching with human bones, particularly for load-bearing applications were addressed. However, the types of polymer composites that suitable for these applications were not mentioned. It is known that the mechanical properties of most polymers are much lower than those of human bones. However, PEEK-based composites, especially reinforced with carbon fibers, and even graphene oxide can meet those mechanical requirements. The authors should mention and add related references in their revised paper

Answer: We thank the reviewer for pointing the need to mentioned the research on the topic of PEEK polymers and composites.

Improvement in the text: Line 473

Among synthetic polymers, poly(aryl-ether-ether-ketone) (PEEK) implants have attracted much attention, for example in cranial, maxillofacial or orthopedic surgeries, e.g. as fusion cages for intervertebral fusion. PEEK is a chemically stable polymer under physiological conditions, with mechanical properties matching those of bone, thus making it suitable for load-bearing application [127]. PEEK-based composites, for example reinforced with carbon fibers and graphene oxide, can also allow increasing mechanical strength to reach clinical requirements. However, PEEK is inherently bioinert. While bioinertness can be advantageous in certain contexts, it limits cell adhesion and increases the risk of fibrous tissue formation around the implant, which may lead to loosening or implant failure. To address these limitations, the surface of PEEK implants, trials to modify PEEK chemically have been attempted [128,129] as well as upon structural modification [130]. In addition, PEEK has also been associated with inorganic phases to improve its biological response either through coating treatment [131] or by mixing it with materials such as SiN or CaP [132,133]. These modifications aim to enhance PEEKs’ integration with the surrounding tissue and cells, and reduce the risk of complication. In contrast to PLA and PGA, which are designed for temporary applications, PEEK-based materials aim to function as long-term implants. Research is still ongoing to increase its bioactivity and osteointegration capacities.

Reviewer 2 Report

Comments and Suggestions for Authors

This article reports a short review on bone healing materials of different types, including metals, ceramics, glasses and polymers and hybrid organic-inorganic bioactive composites. It can be published, but an introduction of recent research on non-oxide ceramics, especially, silicon nitride and related composites should be included. For example,

Giuseppe Pezzotti, Silicon nitride as a biomaterial, Journal of the Ceramic Society of Japan 131 (2023) 398

Hu, G., et al.,. Comparison of surface properties, cell behaviors, bone regeneration and osseointegration between nano tantalum/PEEK composite and nano silicon nitride/PEEK composite. Journal of Biomaterials Science, Polymer Edition, 33 (2022) 35–56.

Neelam Ahuja et al., A Comparative Study on Silicon Nitride, Titanium, and Poly‐Ether Ether Ketone on Mouse Pre‐Osteoblasts Cells, Med Devices Sens, 41 (2021) e10139.

Seunghun S. Lee, et al., Silicon Nitride, a Bioceramic for Bone Tissue Engineering: A Reinforced Cryogel System With Antibiofilm and Osteogenic Effects, Front. Bioeng. Biotechnol., 9 (2021) 794586 

Author Response

Comments 1: An introduction of recent research on non-oxide ceramics, especially, silicon nitride and related composites should be included. For example:

Giuseppe Pezzotti, Silicon nitride as a biomaterial, Journal of the Ceramic Society of Japan 131 (2023) 398

Hu, G., et al.,. Comparison of surface properties, cell behaviors, bone regeneration and osseointegration between nano tantalum/PEEK composite and nano silicon nitride/PEEK composite. Journal of Biomaterials Science, Polymer Edition, 33 (2022) 35–56.

Neelam Ahuja et al., A Comparative Study on Silicon Nitride, Titanium, and Poly‐Ether Ether Ketone on Mouse Pre‐Osteoblasts Cells, Med Devices Sens, 41 (2021) e10139.

Seunghun S. Lee, et al., Silicon Nitride, a Bioceramic for Bone Tissue Engineering: A Reinforced Cryogel System With Antibiofilm and Osteogenic Effects, Front. Bioeng. Biotechnol., 9 (2021) 794586

Answer: The authors thank the reviewer for this comment and the articles references. The main text has been implemented with a section 3.1.4 Non-oxide ceramics. In which the recommended articles are cited along with others.

Improvements in the text: Line 446:

3.1.4 Non-oxide ceramics

Non-oxide ceramics may also be encountered, although less frequently, in the bone repair field. They usually exhibit high strength, fracture toughness, excellent wear resistance and chemical stability [121]. Among them, silicon nitride (SiN) has emerged as a promising material in the field of bone substitutes. In particular, silicon nitride presents high mechanical strength making it suitable for load-bearing application, while its bio-compatibility and ability to promote bone cells activity enhance osteoconductivity [122,123]. In addition, SiN was reported to have an intrinsic antibacterial effect due to its surface chemistry, which inhibits bacterial biofilm formation, hence limiting the risk of infection [123,124]. Finally, the chemical stability of SiN further ensures a long-term performance in physiological environment, which may be relevant for some clinical applications. One of the limitations is however the cost of fabrication of such non oxide ceramics. Also, they are rather complex to produce, which limits their spreading as bone substitute alternatives.

Reviewer 3 Report

Comments and Suggestions for Authors

The authors presented a mini-review and appealed to the topic of bone regeneration.  The  review article is well written and contains the latest findings in the field  and is therefore worthy of publication

the authors should correct on page 7, lines 330 and 331 statement 

Ceramic materials made of HA and tricalcium phosphates (TCP, in the 330 or forms) provide osteoconductive properties relevant to bone repair [93

Author Response

Comments 1: The authors should correct on page 7, lines 330 and 331 statement

Ceramic materials made of HA and tricalcium phosphates (TCP, in the 330 or forms) provide osteoconductive properties relevant to bone repair [93]

Answer: We thank the reviewer for pointing this mistake. The text has been corrected accordingly.

Improvements in the text: Line 348-349:

[…] of HA and tricalcium phosphates (TCP, in the α or β forms) […]

Reviewer 4 Report

Comments and Suggestions for Authors

All authors have same affiliation. No need to mark as 1.

In abstract please highlight the global research trends.

Insert more keywords.

Line 101, check the in-text citations.

There are many problems in bone healing bioactive properties. Please discuss how it can be increased using materials.

Although authors mentioned many materials overview which is lack of in-depth information for the readers.

No figure no table in this manuscript.

No mechanism mentioned in the manuscript. Its random preparation.

Focus specific materials with in depth discussion in this review. In Overview, already many reviews available in this topic.

Comments on the Quality of English Language

Need to improve.

Author Response

Comments 1: All authors have same affiliation. No need to mark as 1.

Answer: Thank you for pointing this mistake. It has been corrected.

Comments 2: In abstract please highlight the global research trends.

Answer: We thank the reviewer for this comment and we added a sentence to address this lack of global trend in the abstract.

Improvements in the text: Line 11 The global trend in bone implant research is shifting toward osteointegrative, bioactive, and possibly stimuli-responsive biomaterials and, where possible, resorbable implants that actively promote the regeneration of natural bone tissue.

Comments 3: Insert more keywords.

Answer: We thank the reviewer for pointing the need of more keywords. XXXX have been added

Improvements in the text: Line 19 Tissue regeneration; Implants; Bone materials

Comments 4: Line 101, check the in-text citations.

Answer: Thank you for pointing this mistake. It has been corrected

Improvements in the text: Line 105 Indeed, bone is a multi-scale composite material that evolves with time and upon mechanical stress and host cells activity [20–24].

Comments 5: There are many problems in bone healing bioactive properties. Please discuss how it can be increased using materials.

Answer: We have now clarified this point and added a portion of text accordingly in the section 2. on page 3. Then each case has been treated in the various subsections that follow.

Improvements in the text:

Line 109:

There are currently various issues relating to bone-related surgical operations and the global recovery of patients. Although not exhaustive, one can mention in particular risks of implant loosening due to poor implant-tissue interaction, low mechanical properties, implant evolution/corrosion in vivo, risks of infections and necrosis if not enough vascularized, slow bone healing, etc. Various types of bone biomaterials have thus been explored – which is still ongoing – to address these points.

Comments 6: Although authors mentioned many materials overview which is lack of in-depth information for the readers.

Answer: The plan of the mini-review was set up to facilitate the reading and view at a glance the different subparts and types of biomaterials encountered in the bone regeneration field. Although it is not possible in a mini-review to extensively develop each part, most relevant references are given in all subsections to allow the Readers to go in further details if needed on the understanding of the different aspects of research; nonetheless, we have highlighted the main trends and hot topics in relation to the development of ever-more biofunctional/bioactive implants, possibly stimuli-responsive and resorbable, which is expected to draw the actual picture of R&D in bone substitutes, and unveil the main paths of innovation.

Comments 7: No figure no table in this manuscript

Answer: Thank you for this comment. We agree with the comment of the Reviewer and add two original illustrations to improve the manuscript.

Improvements in the text: Line 293
A figure has been added with the caption:

Figure 1 overview of the different bone substitute materials categorized by their chemistry, their resorbable or non-resorbable behaviors and shaping possibilities: metals (316L, Ti-3Al-4V, magnesium alloys as bulks solids or porous structures; ceramics (Calcium Phosphate, SiN) as scaffolds, coatings or granules; bioactive glasses as scaffolds or granules; polymers (PEEK, PLA, PLGA…) as scaffolds, coatings or injectable forms.

Line 626:
This evolution is illustrated in the Figure 2

Line 639
A figure has been added with the caption

Figure 2 general evolution in the development of bone substitutes, from the initial need for protection and restoration of the mobility to the latest bone substitute materials that are bioactive and promote bone regeneration by activating and stimulating cell activity in the surrounding of the implant.

Comments 8: No mechanism mentioned in the manuscript. Its random preparation.

Answer: The idea behind this mini-review was not to go into details on all the corresponding mechanisms underlying each subfamily of compound, but rather to give a global insight to Readers new to the field or who desire to see the global picture. In all cases, references have been given where details on characteristic features are given as well as mechanistic schemes when found relevant by these authors. Although mechanisms are not strictly sensu addressed in detail in the present mini-review the main facts are reminded and the main reasons why each family of biomaterial is used and related limitations are highlighted throughout the manuscript.

Comments 9: Focus specific materials with in depth discussion in this review. In Overview, already many reviews available in this topic.

Answer: Although various review reports exist, our interaction with colleagues in the biomaterial field but not specifically in the bone regeneration area led us to identify a global lack of knowledge of the community that is not strictly in this bone-related domain. It was therefore decided, with the Editor of the present journal, to propose this mini-review to draw the present-time global picture of the bone substitute field as of today, and also identify/pin-point the main paths of research and innovation. This was then done per family of biomaterials to yield a short but focused mini-review of relevant to the scientific community in need of rapidly engaging in the bone repair field. Providing an extended and lengthy description of characteristics, properties, mechanistic schemes etc was not the aim of this mini-review, which instead was intended to address a large non-specialist audience to provide a clear insight on R&D and a picture of the bone implants on the market. This mini-review could thus also interest policy-makers, patent agencies, etc to rapidly identify the key paths of research and current trends, to better address clinical needs potentially associated with global population aging and related societal questions.

Round 2

Reviewer 2 Report

Comments and Suggestions for Authors

The manuscript can be published now.

Author Response

Comments 1: The manuscript can be published now.

Answer: We thank the reviewer for this comment and his valuable contribution

Reviewer 4 Report

Comments and Suggestions for Authors

This review is not suitable for publication. It contains no crucial information and is very poorly organized. Many similar reviews have recently been published in good journals.

Comments on the Quality of English Language

Need so much improvement 

Author Response

Comments 1: This review is not suitable for publication. It contains no crucial information and is very poorly organized. Many similar reviews have recently been published in good journals.

Answer: The aim of this mini-review is to provide an overview of the field of bone-substitute materials research, not to go into the details of the actual R&D. Through the different families of materials, we provide a wide variety of examples and point out the pros and the cons of the different families.

The organization of the review is quite logical (fluid), with a first part introducing the tissue of interest and defining the technical specifications that a substitute material should ideally meet, and a main second part introducing the materials that meet these technical specifications.

While we acknowledge that similar reviews have been published, we believe our work contributes to the field in offering a concise, organized summary that bridges the gap between technical details and a broader audience, particularly addressing non-specialist readers seeking a comprehensive entry point in the topic.
We hope this clarification addresses the reviewer’s concerns and highlight the value of this mini-review.

Comments 2: Need so much improvement 

The English level of this review was proof-checked by one of us who spent three years in the US. We believe it conveys sufficient clarity and does not limit the understanding or the readability of the text. Instead, we aimed a fluid reading and gave now illustrative pictures to summarize the ideas. We believe that the present version of the manuscript meets the standards in scientific and format expected for such a mini-review, as was validated by the 3 other reviewers

Can the reviewer give examples of where improvement is needed?
